# Analysis of High-Risk Neuroblastoma Transcriptome Reveals Gene Co-Expression Signatures and Functional Features

**DOI:** 10.3390/biology12091230

**Published:** 2023-09-12

**Authors:** Mónica Leticia Martínez-Pacheco, Enrique Hernández-Lemus, Carmen Mejía

**Affiliations:** 1Faculty of Natural Sciences, Autonomous University of Queretaro, Queretaro 76010, Mexico; monicalmp@iibiomedicas.unam.mx; 2Computational Genomics, National Institute of Genomic Medicine, Mexico City 14610, Mexico; ehernandez@inmegen.gob.mx

**Keywords:** high-risk neuroblastoma, RNA sequencing, bioinformatics, gene co-expression networks, signaling pathways

## Abstract

**Simple Summary:**

Neuroblastoma is a solid cancerous tumor that forms in the nerve cells of children, most commonly in the adrenal glands, which are located on top of both kidneys. This cancer is usually diagnosed after it has spread to other parts of the body in advanced stages of the disease. Consequently, treatment is often a very aggressive combination of chemotherapy and radiotherapy. Also, the survival rate of neuroblastoma is less than 40%, and this is partially explained by the fact that individuals bear genetic features that might limit the response to current treatments, which are the same for all diagnosed children. Therefore, it is necessary to discover a more efficient and less harmful therapy for cancer that can be suitable for all patients, regardless of their heterogeneity. Hence, we aimed to identify the genes whose expression is shared by multiple high-risk neuroblastoma patients, as well as their biological function. We found 104 genes common for 58 individuals that are involved in crucial cellular processes for neuroblastoma development. Our observations propose a list of genes and their biological functions that can be further investigated as possible therapeutic targets for this type of cancer, regardless of the patient’s genetic features.

**Abstract:**

Neuroblastoma represents a neoplastic expansion of neural crest cells in the developing sympathetic nervous system and is childhood’s most common extracranial solid tumor. The heterogeneity of gene expression in different types of cancer is well-documented, and genetic features of neuroblastoma have been described by classification, development stage, malignancy, and progression of tumors. Here, we aim to analyze RNA sequencing datasets, publicly available in the GDC data portal, of neuroblastoma tumor samples from various patients and compare them with normal adrenal gland tissue from the GTEx data portal to elucidate the gene expression profile and regulation networks they share. Our results from the differential expression, weighted correlation network, and functional enrichment analyses that we performed with the count data from neuroblastoma and standard normal gland samples indicate that the analysis of transcriptome data from 58 patients diagnosed with high-risk neuroblastoma shares the expression pattern of 104 genes. More importantly, our analyses identify the co-expression relationship and the role of these genes in multiple biological processes and signaling pathways strongly associated with this disease phenotype. Our approach proposes a group of genes and their biological functions to be further investigated as essential molecules and possible therapeutic targets of neuroblastoma regardless of the etiology of individual tumors.

## 1. Introduction

Neuroblastoma originates from cells of the neural crest or precursors of the sympathetic nervous system. However, it occurs in the adrenal gland, sympathetic ganglia, and paraganglia or along the spinal cord. It is the most common extracranial solid tumor in childhood, found in 8–10% of infants, and is associated with 15–20% of deaths diagnosed with cancer. The event-free survival of high-risk neuroblastomas is less than 40%, so its effective treatment remains challenging [1].

The specific genetic features of neuroblastoma are well-documented at the mutational and chromosomal level. For instance, MYCN amplification in high-risk neuroblastoma patients is associated with poor survival; meanwhile, non-amplified MYCN neuroblastoma with low-stage or even metastases can differentiate into benign subtypes or regress spontaneously by apoptosis. Another group of high-risk neuroblastoma has hemizygous loss of chromosome 11q and is inversely correlated with MYCN; this relationship is found in 70% of all metastatic tumors [2].

Neuroblastoma subtypes are also related to loss of heterozygosity for 1p, which is associated with tumors showing MYCN amplification, and the loss of 11q material, which is associated with a gain of 7q and 3p and a loss of 4p. Typically, the gain of 17q is related to 11q MYCN amplification [3].

MYCN upstream regulators, such as TP53, TERT, ODC1, MCM7, and MDM2, the MYCN downstream targets KIFAP3, OPHN, RGS7, TOP2A, TWIST1, and TYMS [4], the proliferation inductor ALK [5] and microRNAs, such as the miR-17-92 cluster [6,7], and the miR-181 family, miR-542-5p, and miR-628 [6] are also key players in the development of neuroblastoma.

There is less information on the global gene expression of neuroblastoma. Nevertheless, some efforts on microarray and RNA sequencing (RNA-seq) [8,9,10] have provided insights into its behavior at the transcriptome level, demonstrating the genetic features of neuroblastoma that involve the up- and down-regulation of the expression of thousands of genes.

The analysis of RNA-seq data from 30 high-risk neuroblastoma tumors bearing MYCN amplification shows the over-expression of essential genes such as the HAND2, PHOX2B, and GATA3 transcription factors, the proline synthesis regulators ALDH18A1 and PYCR1, and the synaptic vesicle dopamine transporter VMAT2 [10].

Bedoya-Reina, et al. [9] found a high expression of neurotrophic receptor NTRK2, mesenchymal COL1A2, COL6A3, and COL12A1 genes, migratory LAMA3, CLDN11, and DOCK7 genes, and progenitor BCL11A, ERBB3, RTTN, TP63, ASXL3, POU6F2, and SOX6 genes in high-risk neuroblastoma samples. They also determined that the gene expression profile favors cell motility, metastases, DNA repair, and cell cycle disruption.

Transcriptome investigation has also led to the discovery of changes in the expression of non-coding RNAs, splice variants, and novel junctions [8].

However, the relationship between deregulated genes in neuroblastoma and the biological processes and signaling pathways they might impact needs to be better understood. Addressing this issue is indispensable to better understand and treat the disease.

Altogether, the genetic background of neuroblastoma denotes a heterogeneity in the etiology of the disease that must be influencing the poor response of patients to treatment.

Current neuroblastoma therapy involves harsh chemoradiotherapies that generally leave surviving children with lifelong side effects [1]. The discovery of novel therapeutic targets could improve the outcomes of high-risk patients and reduce the burden of sustained complications for surviving patients. Here, we aim to compare RNA sequencing data of neuroblastoma tumor samples from different patients and normal adrenal gland tissues to perform a series of bioinformatic analyses that can lead us to determine the differentially expressed genes in neuroblastoma, as well as elucidate the common gene expression profile for all patients, the gene co-expression networks, and the signaling pathways associated with the neuroblastoma phenotype. With this, we propose essential molecules of neuroblastoma, regardless of the etiology of individual tumors, that can be further investigated as potential therapeutic targets.

## 2. Materials and Methods

### 2.1. Data Retrieval and Preparation

Transcriptome raw count data of neuroblastoma were obtained from the Therapeutically Applicable Research to Generate Effective Treatments—Neuroblastoma (TARGET-NBL) project of the National Cancer Institute’s Genomic Data Commons (GDC) Data Portal website. The TARGET-NBL project (phs000467) uses comprehensive molecular characterization analyses, including RNA-seq outputs, to determine the genetic changes that drive the initiation and progression of high-risk cases (stage 4, when the tumor has metastasized to many locations) of neuroblastoma [11]. We analyzed 1 dataset with 58 patients, according to our study criteria for sample file selection, which were as follows: (1) data category: transcriptome profile, (2) data type: gene expression quantification, (3) experimental strategy: RNA-seq, (4) workflow type: STAR-Counts, and (5) primary site of tumor: adrenal gland. The data were downloaded and prepared using the “TCGAbiolinks” package from Bioconductor [12,13] (Figure 1).

Since the TARGET-NBL project from the GDC Data Portal does not contain normal control samples for the comparison with the tumor samples, the raw count data of normal adrenal gland tissues were downloaded from the National Institutes of Health’s Genotype-Tissue Expression (GTEx) Project website, which is a repository that collects and analyze gene expression data of multiple human tissues from donors. After searching for adrenal gland tissue, we obtained 159 data files for this study. The data were retrieved, prepared, and scaled with the “recount” package from Bioconductor [14,15] (Figure 1).

Data from tumor and normal tissues were merged on a single data matrix. Since neuroblastoma and adrenal gland transcriptome samples come from different data sources, we tested the count matrix for batch effect using the function “ARSyNSeq” of the “NoiSeq” library of R [16,17], which is based on an Anova Simultaneous Component Analysis (ASCA or AnovaSCA) [18]. We observed that samples are neatly clustered by experimental design (and not by batch) on the ASCA Principal Component Analysis (PCA) plot, in which 87% of the variance is explained by the ASCA PC1 and PC2 (Appendix A). Correction by ARSyNSeq causes overfit (Appendix A), as we can denote less defined clustering and a lower variance explained by the PC1 and PC2 (70%). Therefore, we concluded that the batch effect in these samples was negligible and proceeded with downstream analyses.

Transcripts were annotated for GC content, length, biotype, and gene symbol with the “biomaRt” package prior to normalization by GC content, length, and low-count genes with “TCGAbiolinks” [15] (Figure 1).

### 2.2. Exploratory Analyses of the Features of Neuroblastoma and Normal Adrenal Gland Data

To reduce the high dimensional RNA-seq data, summarize the expression of genes across the neuroblastoma and normal adrenal gland samples, and visualize potential associations or patterns of expression between types of tissues, we used the “ComplexHeatmap” package from Bioconductor [19,20] (Figure 1). Raw counts were normalized by the Z-score method (which normalizes every value in a dataset such that the mean of all the values is 0 and the standard deviation is 1) before clustering analysis.

Also, to denote the differences or relationships between the neuroblastoma and normal adrenal gland samples, count data were scaled, centered, and submitted to a principal component analysis (PCA) using the built-in function “prcomp” [21,22,23] (Figure 1) in the programming language R [24]. Results from both analyses were plotted with the “ggplot2” package in R [25].

### 2.3. Analysis of Differentially Expressed Genes (DEGs) between Neuroblastoma and Normal Adrenal Gland Samples

A differential expression analysis (DEA) was performed on the neuroblastoma and normal adrenal gland tissue gene count data matrix, with the “TCGAanalyze_DEA” extension of the “TCGAbiolinks” package [15], to identify the differentially expressed genes (DEGs) in neuroblastoma regarding normal tissue. This extension allows the use of the “limma” package and the “voom” function from Bioconductor [26] methods for matrix data design, normalization, and transformation. The results were visualized on a heatmap, a barplot, and volcano plots created with the “ComplexHeatmap”, “ggplot2”, and “TCGAbiolinks” packages, respectively (Figure 1). For further analyses, we filtered the output of DEG data. We considered as significantly up-regulated those genes with an expression value (log2 fold change (FC)) greater than 1 and a False Discovery Rate (FDR = −log10 *p*-value) lower than 0.01, and significantly down-regulated those genes with an FC lower than −1 and an FDR lower than 0.01.

### 2.4. Weighted Correlation Network Analysis (WGCNA) of DEGs and Identification of Modules Associated with Neuroblastoma

The DEGs obtained by the differential expression analysis were used to carry out a weighted correlation network analysis (WGCNA) and identify interesting co-expressed gene modules (groups) based on the relationship of the expression and the patient’s clinical trait, in this case, tumor condition, with the R “WGCNA” package [27,28] (Figure 1). Samples were clustered to assess the presence of apparent outliers, and we calculated the soft thresholding power β, to which co-expression similarity is raised to calculate adjacency. Then, gene significance and module membership were calculated to relate modules to neuroblastoma. We visualized the results from the significant modules with a hierarchical clustering dendrogram and heatmaps and the correlation between the gene significance and the module membership with scatter plots using the “WGCNA” package. Finally, information on genes in the most significant modules, where genes show the most robust co-expression relationship among them and which co-expression is highly related to neuroblastoma (FDR < 0.01), was extracted for further analyses.

### 2.5. Functional Enrichment Analyses of Genes from the Significantly Correlated Modules

To elucidate the biological implications of the genes in the brown and blue modules, we conducted functional enrichment analyses using the Kyoto Encyclopedia for Genes and Genomes (KEGG) [29,30,31], Gene Ontology (GO) [32,33,34], and REACTOME Pathway databases [35] (Figure 1).

The KEGG pathway enrichment analysis and gene annotation were performed with the “enrichKEGG” function in the R package “clusterProfiler” [36,37] and the “org.Hs.eg.db” package from Bioconductor [38], and the GO analysis for biological processes was conducted using “enrichGO” in the “clusterProfiler” package. The REACTOME pathway enrichment was made with the “enrichPathway” function of the “ReactomePA” library [39].

The “ggplot2” package was used for the bubble plot and heatmap construction, and the signaling pathways were visualized with the “DOSE” library [40].

### 2.6. Data Processing and Statistical Analysis

Data downloading, processing, and analyses were performed in the RStudio integrated development environment [41] for the programming language R [24]. A *p*-value of less than 0.05 and a False Discovery Rate (FDR) (adjusted *p*-value) of less than 0.01 were considered statistically significant in all the analyses.

## 3. Results

### 3.1. Exploration of the Gene Expression Data Features of Neuroblastoma and Normal Adrenal Gland

As shown in Figure 1, we downloaded and prepared 58 neuroblastoma (tumor) RNA-seq datasets, each from a different patient, from the GDC data portal, with 60,660 transcripts, and 159 adrenal gland (normal) datasets from the GTEx data portal, with 58,037 transcripts. Then, we merged the data into a single transcriptome data matrix containing 61,537 transcripts. These were annotated, filtered, and normalized to obtain a matrix with 23,099 genes for both the normal and tumor samples.

In the first place, we wanted to reveal the patterns of gene expression across the neuroblastoma and adrenal gland samples and the relationships between them. Therefore, we conducted a clustering examination and a principal component analysis (PCA) with the 23,099 genes in the matrix. We observed that all 58 neuroblastoma samples are clustered together; this means they have similar behavior, where a group of genes has a high number of counts, and there are other groups with fewer counts. The 159 adrenal gland samples are clustered together, too, showing low counts in the genes highly expressed in neuroblastoma. Also, the patterns displayed by the clustering plot suggest differences in global gene expression between neuroblastoma patients. Overall, there are groups of genes where the number of counts between sample types is not that different (Figure 2a).

The PCA helped us denote that neuroblastoma samples group together at one region of the plot and the normal samples in another. Although tumor samples are aggregated, they show some dispersion in the plot (Figure 2b). That means that the global gene expression pattern in neuroblastoma (tumor) is different from the adrenal gland (normal) samples and that there is a heterogeneity in the gene expression of neuroblastoma from different patients that needs further investigation to elucidate a common profile that the tumor phenotype can explain.

### 3.2. DEGs between Neuroblastoma and Normal Adrenal Gland Samples

The differential expression analysis we performed with the 23,099 genes from the data matrix returned a total of 13,138 DEGs in neuroblastoma regarding the adrenal gland normal samples, of which 6524 are up-regulated (log2 fold change (FC) > 1, False Discovery Rate (FDR) (−log10 *p*-value) < 0.01) and 6614 are down-regulated (FC < −1, FDR < 0.01). Figure 3a shows a heatmap of the hierarchical clustering of the 132 DEGs with FC <= −10 and those with FC >= 10 values. The difference in gene expression between neuroblastoma and normal tissues can be easily observed.

Classifying the DEGs by their gene type, we see the great majority of them (4656 up-regulated and 4029 down-regulated genes) correspond to protein-coding mRNAs, and the rest belong to the long non-coding RNA (lncRNA) (1260 up-regulated, and 880 down-regulated genes), pseudogene (558 up-regulated, and 1660 down-regulated genes), and other types of genes (50 up-regulated, and 45 down-regulated genes) (Figure 3b).

The 13,138 DEGs of neuroblastoma, compared to normal samples, were visualized in volcano plots to show the statistical significance (FDR < 0.01) versus the magnitude of change in gene expression (FC > 1 for up-regulated genes or FC < −1 for down-regulated genes). Figure 3c highlights the symbol of the top 35 up-regulated genes (FC > 1, FDR < 0.01) and the symbol of the top 35 down-regulated genes (FC < −1, FDR < 0.01).

### 3.3. DEG Co-Expression Analysis (WGCNA) and Identification of Gene Modules Associated with Neuroblastoma

We used the DEGs to perform a weighted correlation network analysis (WGCNA) and determine their co-expression relationship, as well as identify those groups (modules or clusters) of co-expressed genes that are related to neuroblastoma phenotype (trait). In our study, value 16 was chosen as the soft thresholding power because it produced a higher similarity with a scale-free network and contributed to gene clustering. The clustering of genes, based on their expression profiles, returned seven modules of DEGs, each named with a unique color (Figure 4a). The correlations between each module and the neuroblastoma trait were calculated, and the results show there are four modules of genes (brown, blue, yellow, and green) that show strong connection (adjacency) (positive correlation values) and are related to the neuroblastoma trait (*p* < 0.05) (Figure 4b). Among these, the brown and blue modules have the highest values (0.88 both, *p*-value < 0.05) (Figure 4b). The connection strengths (adjacencies) of the modules and neuroblastoma phenotypes are shown in Figure 4c, where the red color indicates a high adjacency (positive correlation) and the blue color represents a low adjacency (negative correlation). We detected that the 650 and 704 genes in the brown and blue modules, respectively, exhibit a strong positive correlation with neuroblastoma (corr = 0.88 with *p* < 0.05, and corr = 0.91 with *p* < 0.05, respectively) (Figure 4d,e). Therefore, genes from the brown and blue modules were used for subsequent analyses.

### 3.4. Functional Enrichment Analyses (KEGG, GO, and REACTOME) of DEGs of Modules Associated with Neuroblastoma

We carried out three functional enrichment analyses to identify the most significant genes in each module and elucidate the biological implications of the co-expressed DEGs in the brown and blue modules. First, a signaling pathway enrichment with the KEGG database, which is a repository of the current knowledge on the molecular interaction and gene networks in the context of biological processes, was performed using the 650 and 704 DEGs of the brown and blue modules, respectively. We found 51 significantly enriched pathways, of which the top 20 are shown in Figure 5a for the genes in the brown modules and 15 enriched pathways for the blue module (Figure 5b) (*p*-value < 0.05, FDR < 0.01). Brown module pathways are related to ligand–receptor interactions, synapse signaling, axon guidance, cell adhesion, and insulin secretion. On the other hand, the genes of the blue module are enriched in pathways such as cell cycle, cellular senescence, viral carcinogenesis, DNA replication, and DNA repair mechanisms. The top 20 KEGG pathways for the brown module are enriched with 119 genes (Figure 5c), and the 15 pathways for the blue module are enriched with 130 genes (Figure 5d).

Second, we conducted a REACTOME pathway enrichment, which analyzes, visualizes, and interprets the pathway knowledge to study the same 650 and 704 DEGs from the brown and blue modules, respectively. The results show that in the top 20 enriched pathways, 97 genes (Figure 6c) from the brown module have a significant role (Figure 6a, *p*-value < 0.05, FDR < 0.01) in the neuronal system, chemical synapses, neurotransmitter receptors, potassium channels, and postsynaptic signaling, among others. Regarding the top 20 enriched pathways of the blue module, 138 genes (Figure 6d) are involved in processes such as cell cycle checkpoints, mitosis, DNA replication, chromatid separation and cohesion, and chromosome maintenance (Figure 6b, *p*-value < 0.05, FDR < 0.01).

Third, we took the 650 DEGs of the brown module and the 704 DEGs of the blue module and submitted them to a GO biological process (term) enrichment analysis. We discovered that many biological processes are significantly enriched, and we extracted the genes from the top 20 of each module. Genes in the brown module are enriched in terms such as regulation of membrane potential, axonogenesis, neurotransmitter transport, synaptic vesicle cycle, potassium transport, and others (Figure 7a, *p*-value < 0.05, FDR < 0.01). In contrast, genes of the blue module are enriched in chromosome segregation, DNA replication, mitotic nuclear division, cell cycle, cell cycle checkpoints, etc. (Figure 7b, *p*-value < 0.05, FDR < 0.01). Data show 183 genes involved in the top 20 GO terms of the brown module (Figure 7c) and 188 genes in the top 20 GO terms of the blue module (Figure 7d).

We compared the outputs from KEGG, REACTOME, and GO analyses, and all three databases estimated that there were 42 DEGs in the brown module (Table 1) involved in 80% (sixteen of the top twenty) of the signaling pathways and biological processes detected separately by the enrichments, such as neurotransmitter synapses, axonogenesis, synaptic vesicle cycle, and inflammatory regulation (Figure 8a). Regarding the blue module, we determined 62 DEGs (Table 1) related to 90% (18 of the top 20) of the processes and pathways from separate analyses. These processes are mitosis, DNA replication, cell cycle checkpoints, chromatids and kinetochores separation and cohesion, and chromosome maintenance (Figure 8b).

Altogether, our analyses returned a list of 104 genes (Table 1) whose co-expression relationship can explain the neuroblastoma phenotype in samples from different patients.

### 3.5. Data Availability

The raw data used to conduct the analyses in this study are freely available from the GDC Data Portal and the GTEx Portal.

The article and Appendix A include all datasets and figures generated for this work.

## 4. Discussion

Since less than 40% of high-risk neuroblastoma patients respond to treatment [1], it is crucial to elucidate the common gene expression profile and regulation networks shared by multiple patients, regardless of the etiology of individual tumors, and, therefore, to contribute to the study and discovery of novel therapeutic targets that improve the outcomes and reduce the complications for surviving.

The exploratory analyses performed with the RNA-seq counts from 58 high-risk neuroblastoma and 159 normal adrenal gland tissues revealed exciting features of data. We observed that neuroblastoma samples cluster together, meaning they have similar expression patterns with a large group of genes with a high number of counts and smaller gene groups with lower counts (Figure 2a). Adrenal gland samples are clustered together, too, showing low counts in the genes highly expressed in neuroblastoma (Figure 2a), meaning the expression patterns of these two types of samples are different from each other, which proves the power of transcriptome analysis in detecting the difference in gene expression between tumor and normal samples. We also saw that the behavior of the expression of neuroblastoma may differ between patients (Figure 2a,b), and this can be explained by the individual genetic features of neuroblastoma in each patient and demonstrates the high heterogeneity that we can find in gene expression across tumor samples of this type [42].

The difference in global gene expression patterns of neuroblastoma (tumor) and adrenal gland (normal) tissues was denoted by the principal component analysis (PCA) of data (Figure 2b); this means that the global gene expression pattern in neuroblastoma is different from adrenal gland normal samples. Although they are grouped together in the plot, we also observed that tumor samples exhibit some dispersion (Figure 2b). These results confirmed that there is a heterogeneity in the gene expression of neuroblastoma from different patients that needs subsequent investigation to elucidate a common profile that can be explained by, or associated with, the tumor phenotype.

We performed a differential expression analysis with the 23,099 genes expressed in both tumor and normal samples to address this need. These allowed us to identify 13,138 differentially expressed genes in neuroblastoma, of which 6524 are up-regulated (log2 fold change (FC) > 1, adjusted *p*-value (False Discovery Rate (FDR)) < 0.01) and 6614 are down-regulated (FC < −1, FDR < 0.01) (Figure 3c). The great majority of neuroblastoma DEGs are protein-coding genes (4656 up-regulated, and 4029 down-regulated) (Figure 3b), a result that has been observed for this type of cancer data analysis [9,10] and that encourages our purpose of elucidating protein-coding genes as crucial molecules in neuroblastoma.

We submitted the neuroblastoma DEG population to a weighted correlation network analysis to identify the groups (modules) of genes that bear co-expression relationships and recognize the modules that are associated with the tumor trait (Figure 4a–c). We observed that the brown and blue modules have the highest correlation (corr = 0.88 with *p* < 0.05, and corr = 0.91 with *p* < 0.05, respectively) to neuroblastoma (Figure 4d,e). This means that the co-expression of DEGs in such modules is strongly related to or can explain the neuroblastoma phenotype in samples from different patients, information that had never been acknowledged until this study and that led us to explore the biological function of those genes.

The 650 and 704 DEGs of the brown and blue co-expression modules, respectively, were subjected to a signaling enrichment analysis with the KEGG database. From the 51 significantly enriched pathways (*p*-value < 0.05, FDR < 0.01) for the 119 genes in the brown module (Figure 5c), the top 20 include ligand–receptor interactions, synapse signaling, axon guidance, cell adhesion, and insulin secretion (Figure 5a), which are all known as processes of the nervous system [43,44,45] in which alterations have been related to neuroblastoma development [45,46]. Regarding the 130 DEGs in the blue module (Figure 5d), we observed 15 enriched pathways (*p*-value < 0.05, FDR < 0.01) related to cell cycle, cellular senescence, viral carcinogenesis, DNA replication, and DNA repair mechanisms, among others (Figure 5b) that, when disrupted, are correlated with neuroblastoma [47,48,49,50,51] and other types of cancer [52,53,54,55,56].

Using the same lists of 650 and 704 DEGs, we conducted a pathway analysis with the REACTOME database. We found out that for the top 20 processes of the brown module, 97 genes (Figure 6c) have a significant role (*p*-value < 0.05, FDR < 0.01) in normal nervous system pathways [43,44,45] whose impairment has been related to neuroblastoma [57,58], such as neuronal system, chemical synapses, neurotransmitter receptors, potassium channels, postsynaptic signaling, and others (Figure 6a). Also, for the top 20 enriched pathways of the blue module, we identified a list of 138 genes (Figure 6d) strongly correlated (*p*-value < 0.05, FDR < 0.01) to cell cycle, mitosis, DNA replication, chromatid separation and cohesion, and chromosome maintenance (Figure 6b), which we know are associated with the establishment of this [47,48,49,51,59] and other types of cancer [53,55,60,61] when they are deregulated.

We also performed a GO enrichment analysis for the 650 and 704 DEGs in the brown and blue modules. The results show an impact in multiple biological processes (terms), such as the regulation of membrane potential, axonogenesis, neurotransmitter transport, synaptic vesicle cycle, and potassium transport impacted by 183 genes of the brown module (Figure 7a,c, *p*-value < 0.05, FDR < 0.01). The alteration of these nervous system functions [44,62,63,64] is associated with neuroblastoma phenotype [45,65]. On the other hand, 188 DEGs from the blue module (Figure 7d) are enriched in chromosome segregation, DNA replication, mitotic nuclear division, cell cycle, and cell cycle checkpoints (Figure 7b, *p*-value < 0.05, FDR < 0.01), and impairment is also involved in this [47,48,49,66] and various types of cancer [53,55,61,67].

We compared the results of the KEGG, REACTOME, and GO functional enrichments. We observed that they share 42 co-expressed DEGs (Table 1) and 16 biological processes and pathways for the brown module (Figure 8a). For the blue module, we obtained 62 DEGs (Table 1) in common for the three analyses, and they are related to eighteen processes and signaling pathways (Figure 8b). This means there is an intersection of results from three enrichment analyses that highlight their robustness and statistical significance.

The dysregulation of those biological processes and signaling pathways, such as neurotransmitter synapses, axonogenesis, synaptic vesicle cycle, inflammatory regulation, mitosis, DNA replication, cell cycle checkpoints, chromatids and kinetochores separation and cohesion, and chromosome maintenance, are related to neuroblastoma [45,46,47,48,49,50,51,57,58,59,62,63,64,66], and the majority of them (inflammatory regulation, mitosis, DNA replication, cell cycle checkpoints, and chromosome regulation) are well-known as hallmarks of cancer [68,69,70].

Finally, we performed a differential expression analysis (DEA) with the publicly available RNA-seq data of the Series GSE49711 (498 neuroblastoma samples) [71,72,73] from the Gene Expression Omnibus (GEO) database [74], and our adrenal gland data from GTEx. We observed 5446 DEGs in neuroblastoma, of which 80.4% (4380 DEGs) are present in our list of 13,138 DEGs. Also, 66 (63.5%) of our 104 genes are deregulated in these neuroblastoma samples (Appendix A). These findings endorse our conclusions about the 104 genes explaining the neuroblastoma phenotype.

From our final list of 104 genes (Table 1), 3 of them (MCM4, MCM7, and TERT) have already been found to be deregulated for their expression in high-risk neuroblastoma [75], but the great majority (101 genes) have not.

Therefore, we recognized a novel population of 104 DEGs with co-expression relationships that can explain the neuroblastoma phenotype from different patients, regardless of their individual heterogeneity.

### Scope and Limitations

The present study explored the biological function characterization of neuroblastoma based on gene co-expression networks. Neuroblastoma is an important neurological tumor in children, in particular. However, functional module characterization has remained largely elusive. Tumor heterogeneity, on the one hand, and restricted access to a large number of well-characterized samples, on the other, has partly caused this. We used a somewhat larger (though still relatively small) set of samples (N = 58) for which whole transcriptome analysis (RNASeq) data are available. Gene co-expression network inference in such high-dimensional settings often requires a larger number of samples (for instance, mutual information calculations of gene co-expression networks require N > 100 to attain statistical significance using algorithms such as ARACNe). Since we do not have access to a larger sample space, we resorted to using the WGCNA algorithm that uses Pearson correlation measures (insensible to non-linear correlations) constrained by power law connectivity distribution laws. It is expected that both the large-scale structure and the biological function landscape of neuroblastoma will be further refined by incorporating both larger sample datasets and more rigorous modeling approaches (which, as we have pointed out, go hand in hand). However, we consider that advancing our understanding of the biological foundations of this disease with currently available resources is indeed a worthy endeavor.

## 5. Conclusions

In this study, we analyzed the transcriptome data from 58 patients diagnosed with high-risk neuroblastoma and compared them with data from normal adrenal gland tissue from 159 donors. We also elucidated a gene expression signature common for all neuroblastoma samples. We identified the differentially expressed genes, built their co-expression networks, and determined their biological function.

Our results provide a novel list of 104 genes correlated in their expression. These genes are involved in biological processes and signaling pathways such as neurotransmitter synapses, axonogenesis, synaptic vesicle cycle, inflammatory regulation, mitosis, DNA replication, cell cycle checkpoints, chromatids and kinetochores separation and cohesion, and chromosome maintenance, which have previously been associated with this tumor. Hence, we propose an explanation for the biological processes and signaling pathway alterations observed in high-risk neuroblastoma through the regulatory networks between its DEGs regarding normal tissue instead of simply describing the phenotype in terms of gene expression changes.

We also suggest that these genes are key molecules of neuroblastoma, despite the individual heterogeneity of the patients, and are potential therapeutic targets that need further investigation and validation.

## Figures and Tables

**Figure 1 biology-12-01230-f001:**
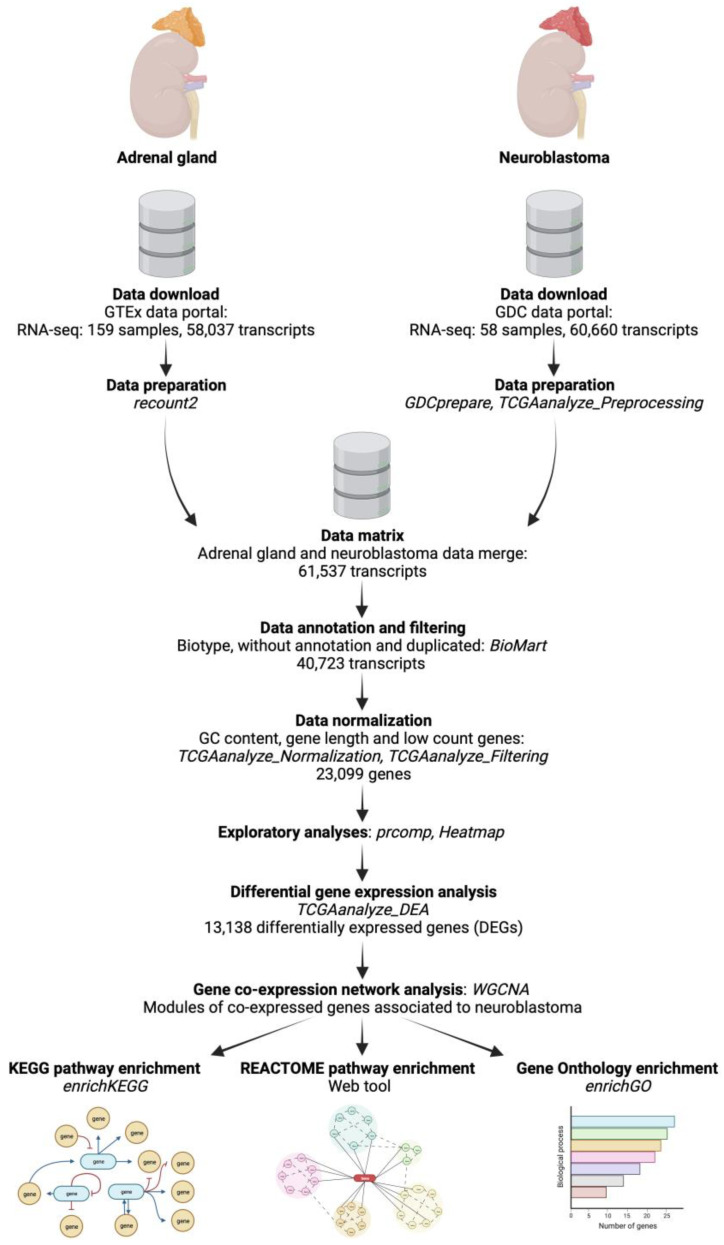
Neuroblastoma transcriptome analysis workflow. A schematic diagram of the whole methodological RNA-seq analysis workflow (created with BioRender.com).

**Figure 2 biology-12-01230-f002:**
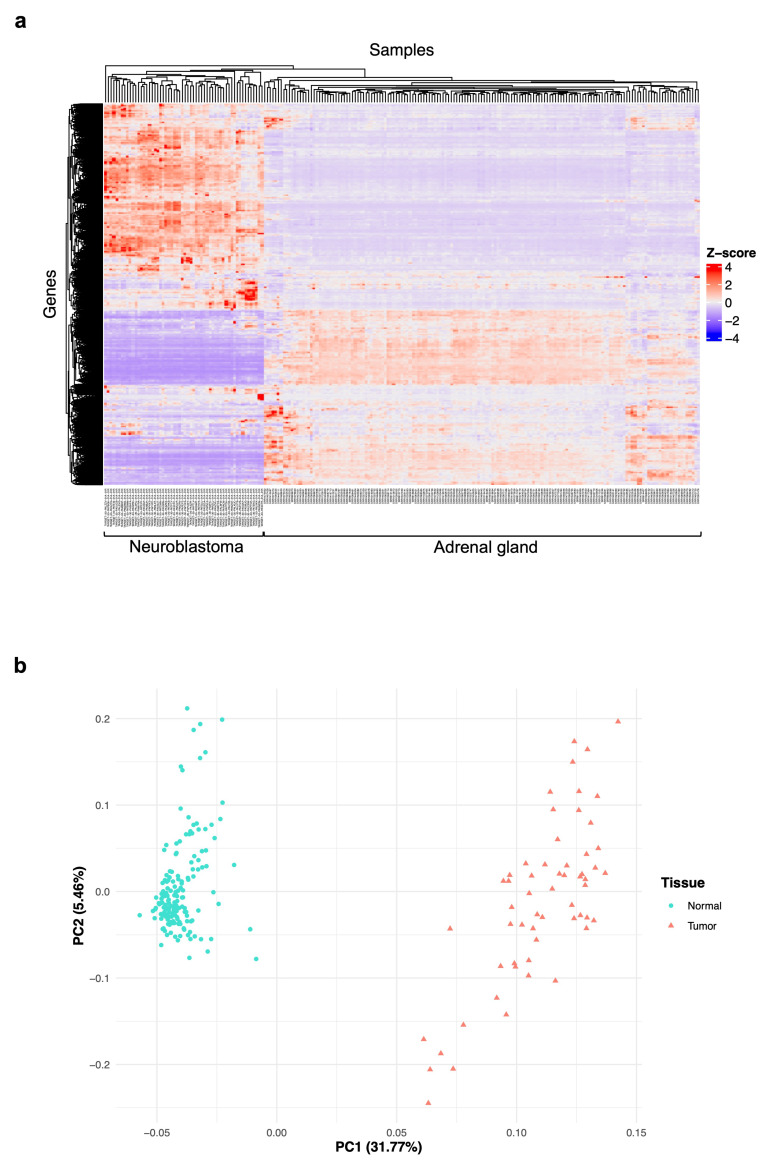
Exploratory analyses of the differences/relationships of gene expression between the neuroblastoma and adrenal gland samples. (**a**) A heatmap displaying hierarchical clustering of the normalized counts for the 23,099 genes in the 217 tissue samples (58 for neuroblastoma (tumor) and 159 for the adrenal gland (normal)). The Z-score represents the value of normalized counts. (**b**) A principal component analysis (PCA) plot of the mRNA expression data based on the top two principal components that characterize the patterns shown by the expression profiles of neuroblastoma and normal tissues. Each dot represents a sample, and each color represents the sample type.

**Figure 3 biology-12-01230-f003:**
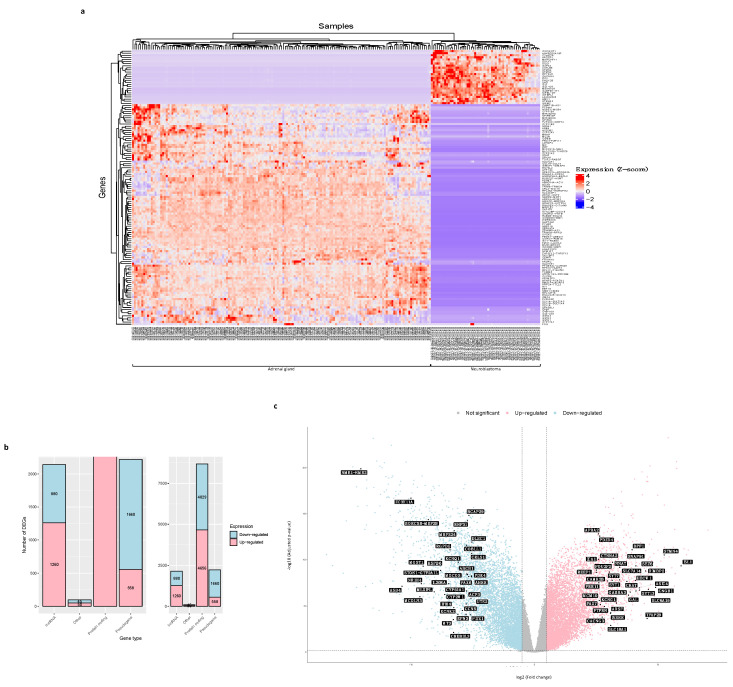
Analysis of differentially expressed genes (DEGs) in neuroblastoma regarding the normal tissue samples. (**a**) A heatmap showing hierarchical clustering of the 132 DEGs in neuroblastoma with expression values lower or equal to −10 and greater or equal to 10. The Z-score represents the gene expression levels (fold change (FC)). (**b**) A bar plot with the classification of DEGs in neuroblastoma by biotype. Pink represents the up-regulated genes and blue represents the down-regulated genes. lncRNA = long non-coding RNA. (**c**) A volcano plot displaying DEGs in neuroblastoma. The pink color represents the up-regulated genes, the blue color represents the down-regulated genes, the grey color represents not-significant genes, and the black labels indicate the top 35 up-regulated genes (n = 6524) and the top 35 down-regulated genes (n = 6614). DEG screening cutoff: FC < −1 and FDR < 0.01. The X-axis represents log2 fold change (FC) and the Y-axis represents −10log (adjusted *p*-value) (False Discovery Rate (FDR)).

**Figure 4 biology-12-01230-f004:**
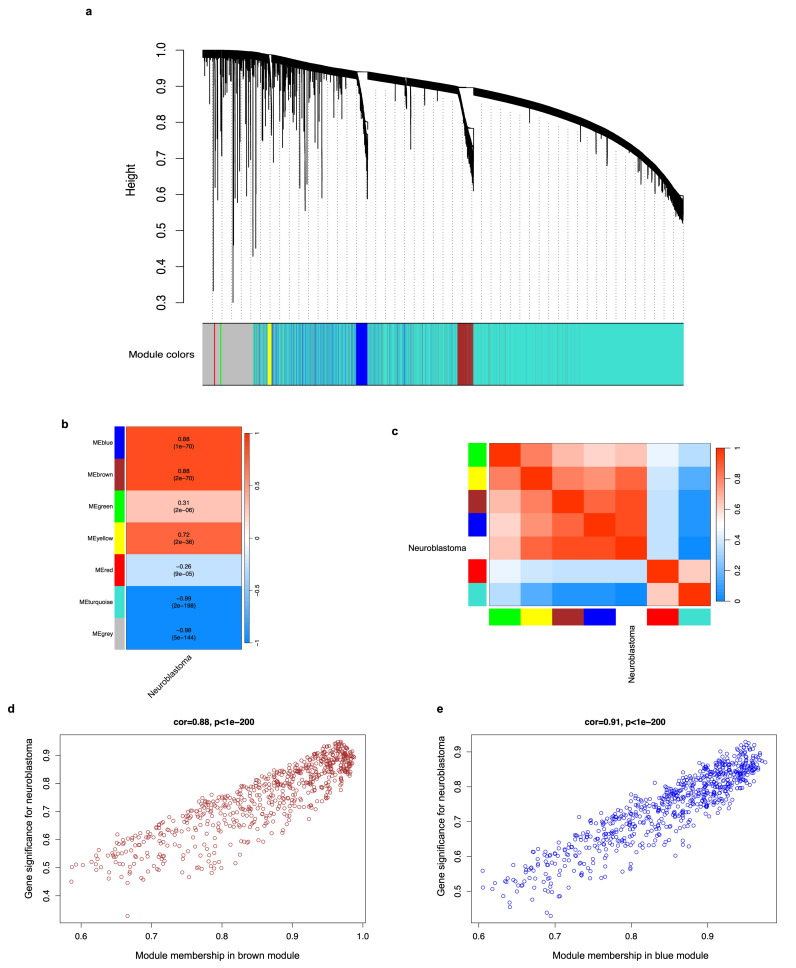
A weighted correlation network analysis (WGCNA) of DEGs and identification of gene modules associated with neuroblastoma. (**a**) Clustering dendrograms of dissimilar DEGs based on the topological overlap and the assigned module (group) colors. Seven co-expression modules were constructed, each with a different color. The tree leaves represent the DEGs, and the height denotes the closeness of individual genes. The grey module gathers the genes with no co-expression relationship. (**b**) A heatmap showing the module-trait (neuroblastoma) relationships. Each row corresponds to expression modules; each cell contains the correlation coefficient and *p*-value. The legend on the right indicates the level of correlation. (**c**) A heatmap displaying the adjacencies (correlation) of gene modules associated with neuroblastoma phenotype. Red represents high adjacency (positive correlation) and blue represents low adjacency (negative correlation). (**d**) A scatter plot of the correlation between the 650 genes of the brown module and neuroblastoma phenotype. (**e**) A scatter plot of the correlation between the 704 genes of the blue module and neuroblastoma phenotype. For all cases, a *p*-value < 0.05 was considered statistically significant.

**Figure 5 biology-12-01230-f005:**
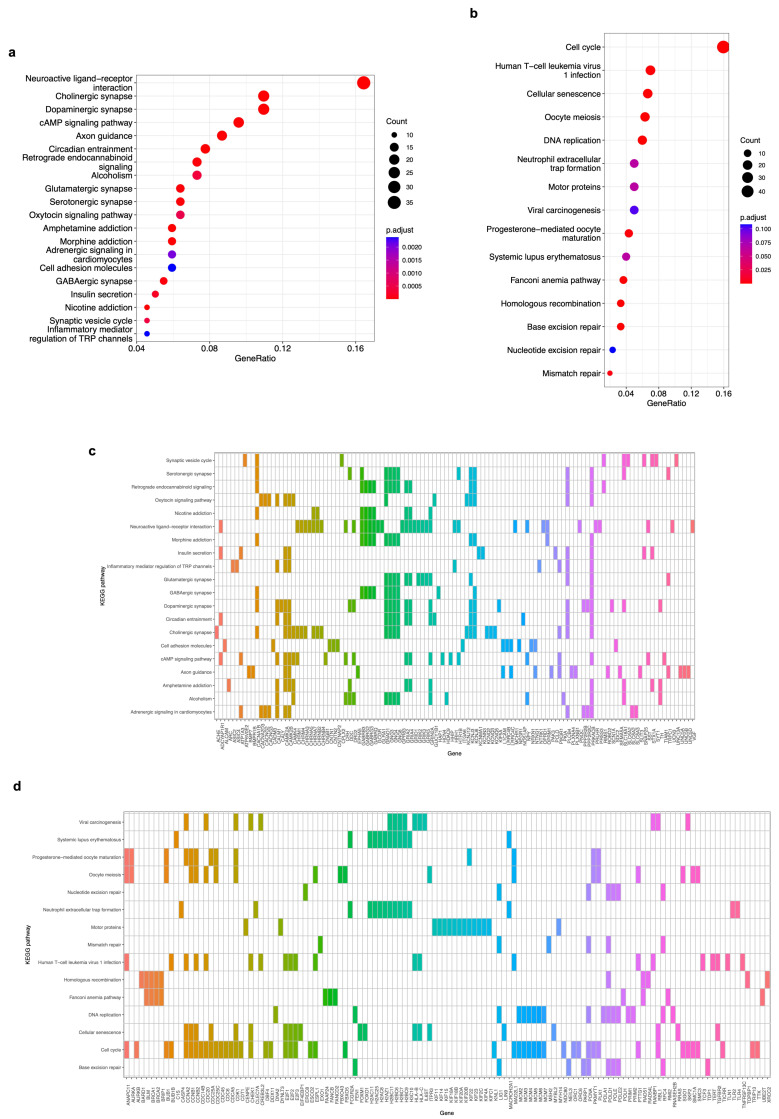
Kyoto Encyclopedia of Genes and Genomes (KEGG) pathway enrichment analysis of genes on the brown and blue modules. (**a**) A bubble plot indicating the enrichment of 119 genes of the brown module in the top 20 KEGG signaling pathways. Labels of the Y-axis show the pathways, and the X-axis corresponds to the gene ratio (number of input genes/number of all genes involved in the pathway). The color of the bubble indicates the enrichment significance, and the size of the bubble represents the number of genes enriched in the pathway. (**b**) Same as in (**a**). The plot displays the enrichment of 130 genes of the blue module in the 15 KEGG signaling pathways. Labels of the Y-axis show the pathways, and the X-axis corresponds to the gene ratio (number of input genes/number of all genes involved in the pathway). The color of the bubble indicates the enrichment significance, and the size of the bubble represents the number of genes enriched in the pathway. (**c**) A heatmap of 119 genes of the brown module that are enriched in the top 20 KEEG signaling pathways. A colored cell indicates that the genes (X-axis) are enriched in the corresponding pathway (Y-axis). (**d**) A heatmap of 130 genes of the blue module that are enriched in the 15 KEEG signaling pathways. A colored cell indicates that the genes (X-axis) are enriched in the corresponding pathway (Y-axis). A *p*-value < 0.05 and an FDR (adjusted *p*-value) < 0.01 were considered statistically significant for all cases.

**Figure 6 biology-12-01230-f006:**
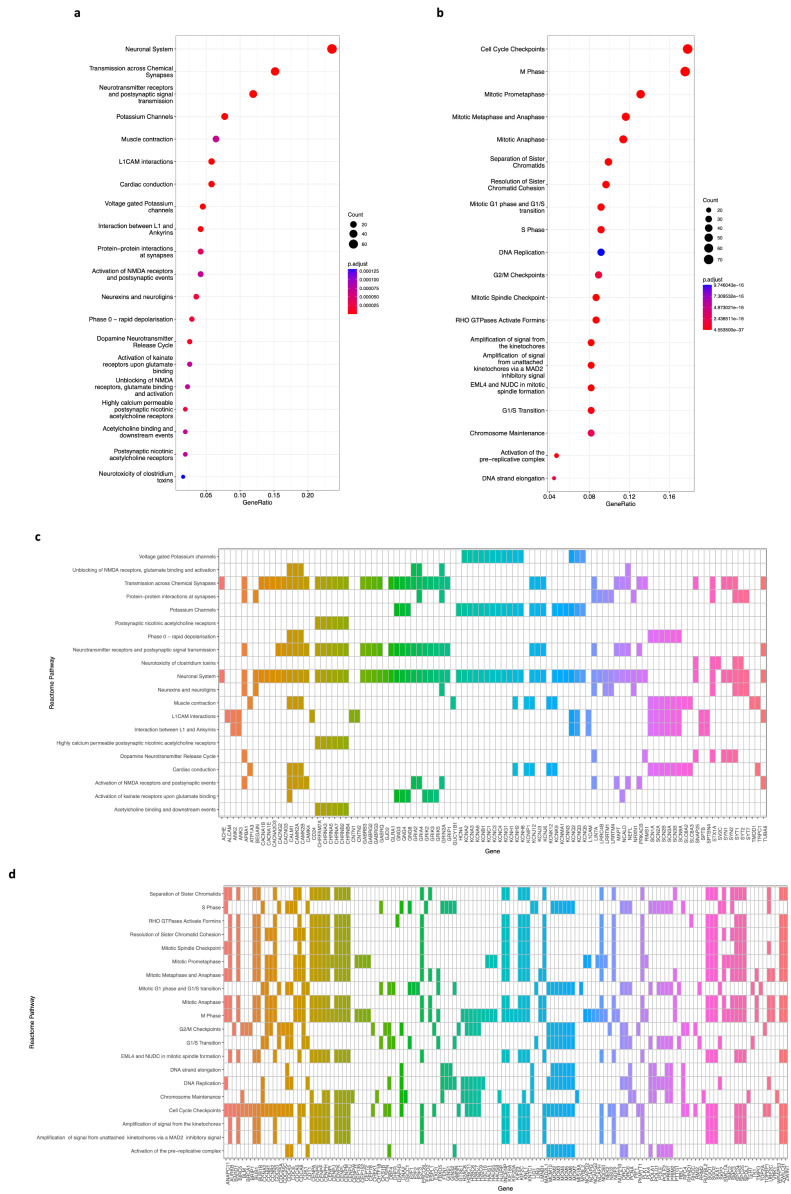
REACTOME pathway enrichment analysis of genes on the brown and blue modules. (**a**) A bubble plot indicating the enrichment of 97 genes of the brown module in the top 20 REACTOME signaling pathways. Labels of the Y-axis show the pathways, and the X-axis corresponds to the gene ratio (number of input genes/number of all genes involved in the pathway). The color of the bubble indicates the enrichment significance, and the size of the bubble represents the number of genes enriched in the pathway. (**b**) Same as in (**a**). The plot displays the enrichment of 138 genes of the blue module in the top 20 REACTOME signaling pathways. Labels of the Y-axis show the pathways, and the X-axis corresponds to the gene ratio (number of input genes/number of all genes involved in the pathway). The color of the bubble indicates the enrichment significance, and the size of the bubble represents the number of genes enriched in the pathway. (**c**) A heatmap of 97 genes of the brown module that are enriched in the top 20 REACTOME signaling pathways. A colored cell indicates that the genes (X-axis) are enriched in the corresponding pathway (Y-axis). (**d**) A heatmap of 138 genes of the blue module that are enriched in the top 20 REACTOME signaling pathways. A colored cell indicates that the genes (X-axis) are enriched in the corresponding pathway (Y-axis). A *p*-value < 0.05 and an FDR (adjusted *p*-value) < 0.01 were considered statistically significant for all cases.

**Figure 7 biology-12-01230-f007:**
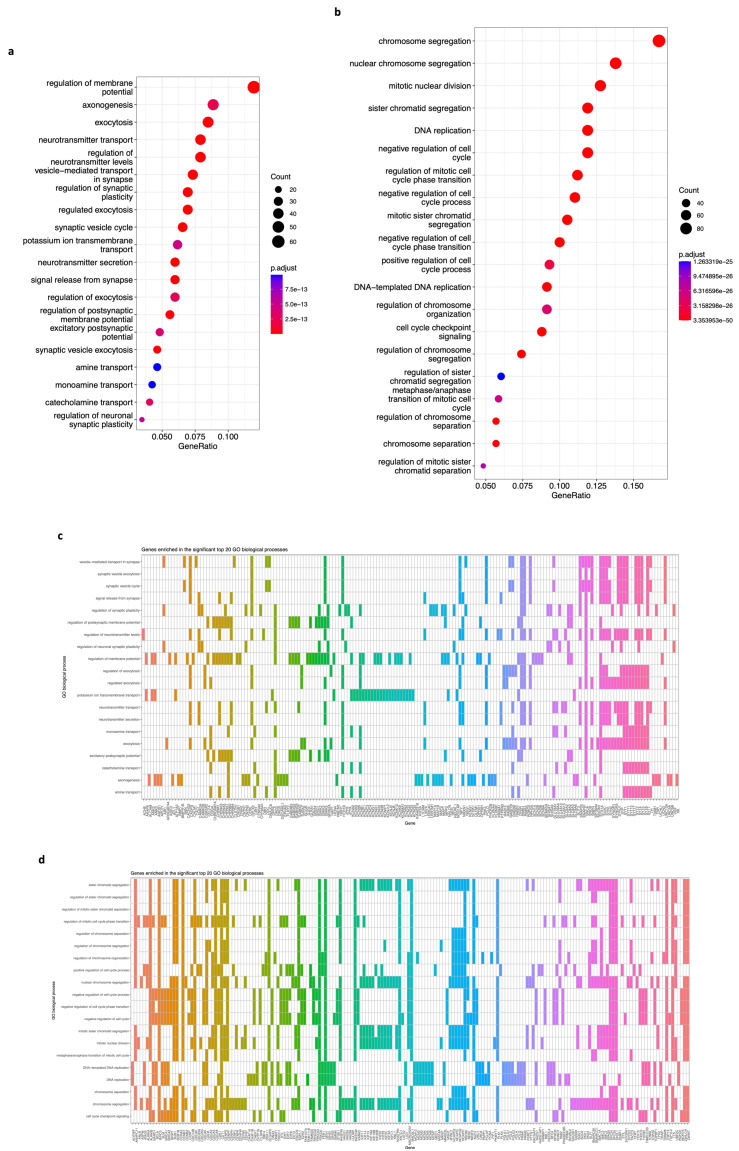
Gene Ontology (GO) biological process (term) enrichment analysis of genes on the brown and blue modules. (**a**) A bubble plot indicating the enrichment of 183 genes of the brown module in the top 20 GO terms. Labels of the Y-axis show the pathways, and the X-axis corresponds to the gene ratio (number of input genes/number of all genes involved in the term). The color of the bubble indicates the enrichment significance, and the size of the bubble represents the number of genes enriched in the GO term. (**b**) Same as in (**a**). The plot displays the enrichment of 188 genes of the blue module in the top 20 GO terms. Labels of the Y-axis show the pathways, and the X-axis corresponds to the gene ratio (number of input genes/number of all genes involved in the term). The color of the bubble indicates the enrichment significance, and the size of the bubble represents the number of genes enriched in the GO term. (**c**) A heatmap of the 183 genes enriched in the top 20 GO terms for the brown module. A colored cell indicates that the genes (X-axis) are enriched in the corresponding term (Y-axis). (**d**) A heatmap of the 188 genes enriched in the top 20 GO terms for the blue module. A colored cell indicates the genes (X-axis) are enriched in the corresponding GO term (Y-axis). A *p*-value < 0.05 and an FDR (adjusted *p*-value) < 0.01 were considered statistically significant for all cases.

**Figure 8 biology-12-01230-f008:**
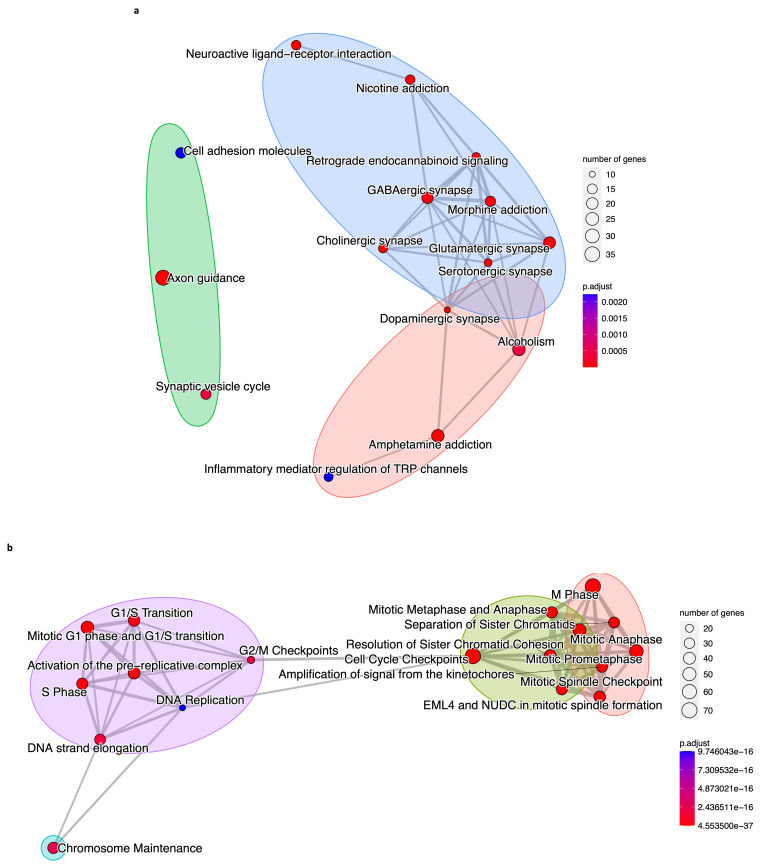
Biological processes and signaling pathways regulated by neuroblastoma’s most significant co-expressed DEGs. (**a**) Biological processes and pathways obtained by the KEGG, REACTOME, and GO functional enrichments are to be regulated by the 42 genes of the brown module. (**b**) Same as in (**a**). Biological processes and pathways obtained by the KEGG, REACTOME, and GO functional enrichments are to be regulated by the 62 genes of the blue module. For (**a**,**b**), the enrichment terms are grouped by category with color circles. A *p*-value < 0.05 and an FDR (adjusted *p*-value) < 0.01 were considered statistically significant for both cases.

**Table 1 biology-12-01230-t001:** Co-expressed DEGs shared by 58 patients diagnosed with high-risk neuroblastoma.

Co-Expression Module	Gene Symbol			
Brown module	ACHE	CHRNB4	GRIK5	L1CAM
	ALCAM	CNTN1	GRIN2A	NRXN1
	ATP1A3	CNTN2	HCN4	RIMS1
	CACNG2	GABRB3	KCNJ12	SCN1A
	CALM1	GABRG2	KCNJ3	SLC8A2
	CAMK2A	GABRG3	KCNJ6	SLC8A3
	CAMK2B	GABRQ	KCNMA1	SNAP25
	CHRNA3	GLRA1	KCNN3	STX1A
	CHRNA5	GRIA4	KCNQ2	SYT1
	CHRNA7	GRIK2	KCNQ3	
	CHRNB2	GRIK3	KCNQ5	
Blue module	ANAPC11	CDCA5	LIG1	POLD1
	AURKB	CDK1	MAD2L1	POLE
	BARD1	CDT1	MCM2	POLE2
	BLM	CENPE	MCM3	PRIM1
	BRCA1	CHEK1	MCM4	PRIM2
	BRIP1	DBF4	MCM5	PTTG1
	BUB1	DNA2	MCM6	RBL1
	BUB1B	E2F1	MCM7	RFC4
	CCNA2	ESCO2	MYBL2	RMI2
	CCNB1	ESPL1	NDC80	SGO1
	CCNB2	FBXO5	ORC1	SMC1A
	CDC20	FEN1	ORC6	SMC3
	CDC25A	KIF18A	PCNA	TERT
	CDC25C	KIF23	PKMYT1	TOPBP1
	CDC45	KIF2C	PLK1	
	CDC6	KNL1	POLA1	

## Data Availability

Publicly available datasets were analyzed in this study. This data can be found here: TARGET-NBL project (phs000467) https://www.cancer.gov/tcga (accessed on 15 May 2023). The data presented in this study are available on request from the corresponding author.

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
