# Peer review of "Analysis of High-Risk Neuroblastoma Transcriptome Reveals Gene Co-Expression Signatures and Functional Features"

_biology, 2023, doi:10.3390/biology12091230_

Round 1

Reviewer 1 Report

The research manuscript is titled " Analysis of high-risk neuroblastoma transcriptome reveals novel gene co-expression signatures and functional features”.

The Introduction provides an adequate background about neuroblastoma and the objectives of the work.

The study's hypothesis is clear; however, I have a couple of minor analysis concerns which can greatly improve the manuscript.

1.    It might be useful to insert a supplementary table of each result included in this manuscript (DEGs, blue and brown module genes expression, pathway enrichment tables, etc.) about the analyzed public data.

2.    I don’t see the human-readable co-expression network file as (Node A; Node B; and r-value), The module information in node attributes is also missing.

3.    Please correct existing typos (eg. Line 151: “TCGAbiolonks”) in the manuscript.

4.    The figures are not appealing, please choose a better layout for the displayed networks.

5.    Materials and methods: lines 153-156: Can the authors clarify the use of FDR or p-value for DEGs consideration?

6.    Figure 3 c and d: Display up-and down-regulated genes in a single plot.

7.    Please add a limitations section of the study.

 Please correct existing typos (eg. Line 151: “TCGAbiolonks”) in the manuscript.

Reviewer 2 Report

Analysis of high-risk neuroblastoma transcriptome reveals novel gene co-expression signatures and functional features.

A brief summary

In this study, the authors investigated transcriptome of neuroblastoma tumor samples from various patients and compare them with normal adrenal gland tissues using RNA sequencing.

They found that 13,138 differentially expressed genes in neuroblastoma, of which 6,524 are up regulated and 6,614 are down-regulated with majority of them are protein-coding genes (4,656 up-regulated, and 4,029 down-regulated).

Specific comments

1. Have you considered the batch effect when you received sequencing data from different sources?

2. Please rewrite below sentences from Material and methods section:

“We considered as significantly up-regulated those genes with 153 an expression value (log2 fold change (FC)) greater than one and a False Discovery Rate 154 (FDR) (-log10 p-value) lower than 0.01, and significantly down-regulated those genes with 155 an expression value (log2 fold change (FC)) lower than -1 and a False Discovery Rate 156 (FDR) (-log10 p-value) lower than 0.01”.

3. In the discussion section, please give more explanation about link of GO function of identified genes and carcinogenesis.

4. Explain more and in detail in the discussion section, about link of Reactome results of function of identified genes and carcinogenesis.

NA

Reviewer 3 Report

1.     Could the authors provide the IRB number?

2.     If datasets from a platform were used, did the authors request permission for its use?

a.     If datasets were used, the abstract should be rewritten to avoid misunderstanding.

b.     If datasets were used, the title should be revised to reflect the study's methodology.

3.     The authors wrote about data availability L393-396. Could the authors upload the dataset used by them?

Reviewer 4 Report

Tumor heterogeneity is a common feature among neuroblastoma patients. Mónica Leticia utilized a WGCNA-based approach to identify shared differentially expressed genes and their biological functions in high-risk neuroblastoma patients through analysis of publicly available bulk RNA-seq databases. These genes were explored as potential therapeutic targets for neuroblastoma. However, I do have some points of concern to raise below and some queries where data could be included to improve the readability of the manuscript.

In Line 110: “We analyzed 58 datasets, each from a different patient, according to our study criteria for sample file selection”. The description might lead to confusion. Changing it to "one dataset with 58 patients" could be clearer.

In Line 215:  ”Also, we can notice different expression profiles between neuroblastoma pa-215 tients“. The author should provide clearer details about which specific genes or features these distinct expression profiles refer to.

In Line 240: Consider integrating Figures 3c and 3d into a single image to enhance clarity.

Regarding batch effects, given that the tumor data is from TARGET and normal tissue data is from the National Institutes of Health's Geno-118 type-Tissue Expression (GTEx) Project website, did the authors account for batch effects during analysis?

Why were the yellow and green modules discarded during the WGCNA analysis?

In the pathway analysis, why did the authors use differentially expressed genes from KEGG for GO analysis and use enriched genes from GO for REACTOME analysis? What's the rationale and benefit of this approach? Does the order of KEGG, GO, and REACTOME datasets effect the result?

How can the 103 genes derived from REACTOME analysis explain the neuroblastoma phenotype across various patient samples? Have the authors validated these findings on other databases, such as GSE49711?

Minor editing of English language required
